# Viral Vectors for COVID-19 Vaccine Development

**DOI:** 10.3390/v13020317

**Published:** 2021-02-19

**Authors:** Kenneth Lundstrom

**Affiliations:** PanTherapeutics, CH1095 Lutry, Switzerland; lundstromkenneth@gmail.com

**Keywords:** SARS-CoV-2, COVID-19, vaccines, adenovirus, preclinical immunization, clinical trials, approved vaccine

## Abstract

Vaccine development against SARS-CoV-2 has been fierce due to the devastating COVID-19 pandemic and has included all potential approaches for providing the global community with safe and efficient vaccine candidates in the shortest possible timeframe. Viral vectors have played a central role especially using adenovirus-based vectors. Additionally, other viral vectors based on vaccinia viruses, measles viruses, rhabdoviruses, influenza viruses and lentiviruses have been subjected to vaccine development. Self-amplifying RNA virus vectors have been utilized for lipid nanoparticle-based delivery of RNA as COVID-19 vaccines. Several adenovirus-based vaccine candidates have elicited strong immune responses in immunized animals and protection against challenges in mice and primates has been achieved. Moreover, adenovirus-based vaccine candidates have been subjected to phase I to III clinical trials. Recently, the simian adenovirus-based ChAdOx1 vector expressing the SARS-CoV-2 S spike protein was approved for use in humans in the UK.

## 1. Introduction

Severe acute respiratory syndrome coronavirus 2 (SARS-CoV-2) has spread quickly around the world, causing the COVID-19 pandemic, which has seen more than 100 million infections, 2.15 million deaths and a severely damaged global economy [1]. The severity and spread of COVID-19 were unprecedented compared to previous coronavirus outbreaks for SARS-CoV in 2004–2005 [2] and Middle East Respiratory Coronavirus (MERS-CoV) in 2010 [3]. As there were no efficient antiviral drugs or existing vaccines against SARS-CoV-2 when the pandemic started, a frantic drug and vaccine development process was initiated. This involved repurposing existing available antiviral drugs and vaccine development based on all possible technologies including inactivated and attenuated viruses, protein- and peptide-based vaccines, nucleic acid vaccines and not least, viral vector-based vaccines [4]. Although significant progress has been seen for many types of vaccines, including the approval of two RNA-based vaccines [5], the focus in this review is on viral vector-based vaccines. After the introduction of available viral vector systems, examples of preclinical studies for viral vector-based vaccine studies will be described, followed by a summary of clinical trials conducted or in progress. Finally, conclusions are drawn on what has been achieved so far and what is expected in the future in relation to current and potential future mutations and variations of coronaviruses and how we can prepare for emerging outbreaks.

## 2. Selection of Viral Vector

### 2.1. Adenovirus

Adenoviruses are by far the most utilized and advanced viral vectors developed for SARS-CoV-2 vaccines. They are non-enveloped double-stranded DNA (dsDNA) viruses with a packaging capacity of up to 7.5 kb of foreign genes [6]. In almost all cases, replication-deficient third generation Ad vectors with deletions in the E1 and E3 genes have been engineered for the expression of the SARS-CoV-2 spike (S) protein or epitopes of it such as the receptor binding domain (RBD) [7] (Figure 1). Typically, adenovirus particles are propagated in human embryonic kidney 293 (HEK293 cells) in bioreactors for research and GMP applications [8].

### 2.2. Poxvirus

Poxviruses are large dsDNA viruses with an impressive packaging capacity of more than 30 kb of foreign genetic information [9,10] (Figure 2). Moreover, vaccinia virus vectors have frequently been used for vaccine development [11]. The Modified Vaccinia virus Ankara (MVA) serves as a potent vector system, which has demonstrated immunogenic efficacy against cancer and infectious diseases [12]. MVA vectors have proven extremely safe and have demonstrated protection against challenges with different infectious pathogens in preclinical models. Promising results have also been received from clinical trials on avian influenza virus H1N1 and Ebola virus vaccine candidates.

### 2.3. Lentivirus

Lentiviruses belong to the family of ssRNA retroviruses with similar properties to classic retroviruses, such as packaging capacity up to 8 kb, reverse transcription of RNA to cDNA and integration into the host genome [13,14] (Figure 3). However, in contrast to classic retroviruses, lentiviruses are capable of transducing both dividing and non-dividing cells [13]. Lentivirus vector production has been facilitated by the engineering of packaging cell lines and HEK293 suspension cultures in bioreactors [15]. Furthermore, both ecotropic [16] and pantropic [17] lentivirus vectors have been engineered. A common approach is to generate pseudotyped lentivirus vectors with an envelope structure from vesicular stomatitis virus glycoprotein (VSV-G), which alters the lentivirus tropism and enhances safety [18]. Inducible expression has been established by the engineering of a lentivirus with a tetracycline (Tet) promoter, which can be activated by administration of doxocycline in small cell lung cancer cell lines [19].

### 2.4. Measles Virus

Measles viruses (MVs) are enveloped viruses with an ssRNA genome of negative polarity [20]. They possess a packaging capacity of 6 kb and have the capacity of cytoplasmic self-amplification of RNA. However, the minus strand genome has required the engineering of reverse genetics rescue systems from cloned MV DNA in helper cell lines [21,22]. Generally, the gene of interest has been introduced, either between the phosphoprotein (P) and the matrix protein (M) or between the hemagglutinin (HA) and the large (L) protein (Figure 4A).

### 2.5. Rhabdovirus

Similar to MV, rhabdoviruses possess an ssRNA genome of negative polarity with a packaging capacity of 6 kb [23]. Expression vectors have been engineered for both rabies virus [24] and VSV [25]. Generally, transgene expression from rabies virus (RABV) vectors has been achieved by introduction of the gene of interest between the RABV N and P genes [26] (Figure 4B). In the case of VSV, the gene of interest is introduced between the G and L genes or M and L genes [27] (Figure 4C). The negative polarity of the ssRNA genome of rhabdoviruses has required the engineering of vaccinia virus-base reverse genetics systems [25]. However, vaccinia virus-free production systems have been developed for both RABV [28] and VSV [29].

### 2.6. Alphavirus

Alphaviruses are self-amplifying RNA viruses, which possess an envelope structure and contain a positive strand ssRNA with a packaging capacity of 8 kb [30]. Typically, expression vectors based on Semliki Forest virus (SFV) [30], Sindbis virus (SIN) [31] and Venezuelan equine encephalitis (VEE) [32] have been engineered. Alphavirus vectors have been utilized as recombinant particles, naked RNA replicons and plasmid DNA-based replicons (Figure 5).

## 3. Preclinical Studies

Viral vectors have previously been frequently applied for vaccine development against both viral infections and cancers [20,33]. For example, VSV vectors expressing the Lassa virus glycoprotein (LASV-GPC) provided protection against LASV challenges in immunized guinea pigs and macaques [34]. Moreover, immunization of mice with VSV vectors expressing the Zika virus (ZIKV) membrane-envelope (ME) protein resulted in protection against lethal challenges with ZIKV [35]. Expression of Ebola virus glycoprotein (EBOV-GP) from an adenovirus type 5 (Ad5) vector demonstrated protection against EBOV challenges in primates [36]. VSV-based EBOV-GP expression also showed good protection against Ebola virus disease (EVD) in a phase III clinical trial in Guinea [37]. Additionally, the FDA approved the EBOV vaccine based on the VSV-ZEBOV vector under the brand name Ervebo in December 2019 [38].

In the context of viral vector-based vaccine candidates against SARS-CoV-2, a number of preclinical studies have been conducted (Table 1). Among the 162 on-going COVID-19 vaccine candidate studies listed by the WHO [38], 35 utilizes viral vectors or virus-like particles (VLPs), so that only selected examples are presented here and in Table 1. Adenovirus-based vectors have dominated the field including human Ad5 and Ad26 and the simian AdChOx1. For instance, in a preclinical study, the intramuscularly administered Ad5 vector was utilized for the expression of the codon-optimized SARS-CoV-2 spike protein (S) in rhesus macaques [7]. The immunization with Ad5-S-nb2 elicited systemic S-specific antibody and cell-mediated immune (CMI) responses. Similarly, intranasal injections generated systemic and pulmonary responses although weaker CMI responses than seen for intramuscular administration. Moreover, both intramuscular and intranasal administration provided protection of macaques against challenges with SARS-CoV-2. Additional studies confirmed that immunization with a 10-fold lower dose (1 × 10^10^ Ad5-S-nb2 particles) also conferred protection. In another approach, Ad5 vectors were engineered for the expression of the full-length SARS-CoV-2 S protein, the S1 domain and the RBD [39]. Intranasal immunization of mice with Ad5 expressing the SARS-CoV-2 S RBD (AdCOVID) elicited strong and focused RBD-specific induction of mucosal IgA, serum neutralizing antibodies and CD4^+^ and CD8^+^ T cells with Th1-like cytokine profiles.

A COVID-19 vaccine candidate has also been developed based on Ad26 [40]. A single administration of the Ad26.COV2.S vaccine candidate elicited neutralizing antibodies and protected immunized hamsters against SARS-CoV-2 associated pneumonia and death [40]. Likewise, strong neutralizing antibody responses and protection against challenges with SARC-CoV-2 were achieved in immunized macaques [41]. The Russian Sputnik V vaccine candidate, based on a prime-vaccination with Ad26-SARS-CoV-2 S followed by a booster vaccination with Ad5-SARS-CoV-2 S, showed good safety and immunogenicity profiles in preclinical studies [42].

The full-length SARS-CoV-2 S protein has also been expressed from the chimpanzee Ad vector ChAdOx1 (ChAdOx1 nCoV-19). Immunization of mice and rhesus macaques with ChAdOx1 nCoV-19 elicited robust humoral and cellular responses [43]. Protection against SARS-CoV-2 challenges were obtained in immunized macaques [43]. In another study, the chimpanzee adenovirus was employed for the expression of a prefusion-stabilized spike protein (ChAd-SARS-CoV-2-S), showing robust systemic humoral and CMI responses in mice receiving intramuscular injections [44]. Although protection against lung infection, inflammation and pathology was achieved, viral RNA was still detected. In contrast, intranasal immunization elicited strong neutralizing antibody responses and promoted systemic and mucosal IgA and T cell responses and almost completely prevented SARS-CoV-2 infections in both the upper and lower respiratory tracts.

In another approach, a lentivirus vector expressing the SARS-CoV-2 S protein was systemically administered to ACE2 humanized mice [45]. Immunization experiments showed only partial protection against SARS-CoV-2, despite the presence of robust antibody responses in the serum. In contrast, intranasal administration resulted in immune responses in the respiratory tract, leading to a >3 log10 decrease in viral loads in the lung and reduced local inflammation. Furthermore, both integrative and non-integrative lentivirus vectors prevented lung deleterious injury in golden hamsters. Another intranasal approach relates to the engineering of an influenza virus-based COVID-19 vaccine expressing a membrane-anchored form of the RBD of the S protein replacing the neuraminidase (NA) gene [55]. A single intranasal dose of the ΔNA(RBD)-Flu elicited robust neutralizing antibody responses against SARS-CoV-2 at levels comparable to those observed in COVID-19 patients. Additional studies on intranasal administration of influenza virus-based vaccine candidates are in progress at the National Research Center in Egypt, Hong Kong University, and Institute Butantan in Brazil [56].

Poxviruses and particularly modified vaccinia virus Ankara (MVA) strain have previously demonstrated to provide protective immunity against influenza virus H5N1 in mice [11]. The MVA vaccinia virus strain has also been considered as a promising candidate for vaccine development against coronaviruses, due to their potential targeting of mucosal surfaces of the respiratory tract [57]. A synthetic MVA-based vaccine platform has been engineered by application of a unique three-plasmid system for co-expression of SARS-CoV-2 S and nucleocapsid proteins [46]. Immunization of mice with fully synthetic MVA (sMVA) vectors induced strong SARS-CoV-2 antigen-specific humoral and cellular immune responses. In another study, the full-length SARS-CoV-2 S protein was expressed from an MVA vector and evaluated for immunogenicity in mice [47]. Application of both a DNA/MVA or MVA/MVA prime-boost strategy elicited strong and broad SARS-CoV-2 S-specific CD4^+^ and CD8^+^ T-cell responses. Moreover, both approaches induced high titer IgG antibody responses for the SARS-CoV-2 S protein, including the RBD region. It was also demonstrated that one or two doses of MVA-COV2-S protected humanized K18-hACE2 mice from challenges with SARS-CoV-2 and prevented viral replication in the lungs.

A chimeric Newcastle disease virus (NDV) expressing the SARS.CoV-2 S protein showed stable expression of the membrane anchored S protein [48]. The NDV SARS-CoV-2 vaccine candidate demonstrated strong binding and neutralizing antibody responses in immunized mice and hamsters and protected the animals from SARS-CoV-2 infections. In another study, both the wildtype and a membrane-anchored SARS-CoV-2 S protein were expressed from an NDV vector eliciting high levels of neutralizing antibodies in intramuscularly immunized mice [49]. Both vaccine candidates provided protection against mouse-adapted SARS-CoV-2 challenges.

Among self-replicating RNA viruses, MV vectors have been engineered for the expression of the full-length SARS-CoV-2 protein from two positions in the MV genome [50] (Figure 4A). Mice immunized twice with MV-SARS-CoV-2-S vectors elicited efficient Th1-biased antibody and T cell responses. In the case of a replication-competent VSV-based vector expressing the SARS-CoV-2 S protein, neutralizing antibodies were generated in immunized mice, which provided protection from SARS-CoV-2 related pathogenesis [51]. Moreover, in another approach, the VSV G protein was replaced by the SARS-CoV-2 S protein in a replication-competent VSV-ΔG vector, which elicited potent SARS-CoV-2-specific neutralizing antibodies in immunized golden Syrian hamsters [52]. A single dose of 5 × 10^6^ pfu of VSV-ΔG protected immunized hamsters against SARS-CoV-2 challenges, without causing any significant lung damage and showing no presence of viral load.

Although not directly based on viral vector delivery, the approach of utilizing RNA-based self-amplifying alphavirus vaccine candidates is worth mentioning here. In this context, the VEE-based replicon RNA carrying the prefusion-stabilized SARS-CoV-2-S RNA was encapsulated in lipid nanoparticles (LNPs), which were intramuscularly administered into BALB/c mice [53]. Immunization resulted in high and dose-dependent SARS-CoV-2 specific antibody responses and neutralization of virus. The magnitude of antibody response was higher than seen in recovered CIVID-19 patients, the immunization induced Th1-biased responses, and no antibody-dependent enhancement (ADE) was detected. Moreover, another approach is comprised of virus-like particles (eVLPs) encoding the murine leukemia virus (MLV) Gag protein resulting in budding particles applicable as vaccine candidates [58]. In this context, the trivalent pan-coronavirus vaccine candidate VBI-2901 expressing the SARS-CoV, MERS-CoV and SARS-CoV-2 spike proteins and the VBI-2902 with the monovalent SARS-CoV-2 S have demonstrated robust immunogenicity and efficacy in hamsters [54]. The encouraging preclinical results have supported the initiation of a phase I/II study in early 2021.

## 4. Clinical Trials

According to the WHO, 52 vaccine candidates are currently in progress, of which 14 are based on viral vectors [56]. Examples of viral vector-based clinical trials are presented here and summarized in Table 2. The first-in-human phase I dose-escalation study was conducted in 108 healthy volunteers [59]. Administration of 5 × 10^10^, 1 × 10^11^ and 1.5 × 10^11^ Ad5-SARS-CoV-2 S particles showed good safety and tolerability profiles and elicited T cell and humoral responses. Next, 603 healthy volunteers were subjected to a randomized, double-blind, placebo-controlled phase II trial with two doses of 5 × 10^10^ and 1 × 10^11^ virus particles of the Ad5-SARS-CoV-2 S vaccine candidate [60]. Strong neutralizing antibodies were induced after a single vaccination and the overall safety was good. Additionally, the recruitment of 40,000 healthy adults 18 years or older is in progress for a global double-blind, placebo-controlled phase III trial, where a single dose of 5 × 10^10^ of Ad5-SARS-CoV-2 S is intramuscularly administered [61]. Another phase III study has started in 500 healthy 18–55 years old volunteers, applying a single immunization with a dose of 5 × 10^10^ of Ad5-SARS-CoV-2 S [62].

An Ad26-based vaccine candidate, Ad26.COV2 S, was subjected to a single administration of 1 × 10^10^ or 5 × 10^10^ Ad26.COV2 S particles in a randomized, double-blind, placebo-controlled phase I/II trial in 1045 healthy volunteers in Belgium and the USA [63]. The vaccination was safe and elicited strong immune responses according to interim results [64]. Currently, recruitment for a randomized, double-blind, placebo-controlled phase III trial with 60,000 participants is in progress [65]. Currently, volunteers are recruited for another phase III trial, where two injections of Ad26.COV2 S will be received two months apart [66]. In another clinical approach, the Russian Sputnik V vaccine candidate rAd26-S/rAd5-S showed good safety and only mild adverse events in a phase I/II trial [42]. The strategy of prime immunization with the Ad26 serotype followed by a boost vaccination with Ad5 21 days later intended to overcome pre-existing adenovirus in the population. Intramuscular administration generated robust SARS-CoV-2-specific immune responses in all vaccinees. These positive results supported two randomized, double-blind, placebo-controlled, multi-center phase III trials in healthy volunteers, of which one is recruiting volunteers [67] and the other one has already started [68]. Another randomized, double-blind, placebo-controlled phase III trial in 2000 participants is planned in Venezuela for the rAd26-S/rAd5-S vaccine [69]. Very recently, interim results from a phase III trial with the rAd26-S/rAd5-S vaccine (Gam-COVID-Vac) demonstrated good tolerability and 91.6% efficacy against COVID-19 [70]. Surprisingly, the Sputnik V vaccine was approved in Russia after a preliminary evaluation in 76 volunteers and before any phase III clinical trial had been finished [71].

The most advanced vaccine candidate is based on the simian ChAdOx1 vector expressing the full-length SARS-CoV-2 S protein (ChAdOx1 nCoV-19) [42]. Preliminary results from a phase I/II clinical trial showed good safety and immune responses in 32 out of 35 vaccinated individuals [72]. Furthermore, a booster immunization elicited both humoral and cellular immune responses in all vaccinated individuals. In another single-blind, randomized, controlled, phase II/III trial 100 participants aged 18–55 years 120 aged 56–69 years and 240 aged 70 years and older received intramuscularly 2.2 × 10^10^ particles of the ChAdOx1 nCoV-19 vaccine candidate while individuals in the control group were vaccinated with the meningococcal MenACWY vaccine [73]. An interim report from the phase II part showed local and systemic reactions, such as injection-site pain, feeling feverish, muscle ache and headache, which were more common for ChAdOx1 nCoV-19 than the control vaccine [74]. The adverse events were more common in younger adults than in older adults (aged >56 years). The neutralizing antibody titers were similar for all age groups after two immunizations. In the randomized, double-blind, placebo-controlled multicenter phase III trial, the recruitment of 30,000 participants for the determination of safety, efficacy and immunogenicity is on-going [75]. Moreover, other phase III trials are recruiting or in progress in Russia [76], in the UK [77], and India [78]. Recently, the ChAdO1 nCoV-19 vaccine was approved by the regulatory authorities in the UK [79]. Interim results from four randomized phase III trials in the UK, Brazil and South Africa demonstrated an acceptable safety profile and showed 62.1% efficacy in individuals who received two standard doses of 5 × 10^10^ ChAdOx1 nCoV-19 particles and 90.0% after administration of a low dose of 2.2 × 10^10^ particles, followed by a standard dose [80].

In addition to adenovirus vectors, the vaccinia virus MVA strain has been engineered for the expressionm of the SARS-CoV-2 S protein [46,47] and volunteers are now recruited for the first-in-human phase I clinical trial in healthy volunteers [81]. Safety and tolerability will be evaluated at two doses of 1 × 10^7^ and 1 × 10^8^ IU of the MVA-SARS-COV-2 S vaccine candidate. In another vaccinia virus-based approach, the synthetic MVA expressing various regions of the SARS-CoV-2 genome will be evaluated for safety and tolerability in 129 healthy volunteers in a phase I trial [82]. Volunteers are currently recruited for vaccinations with 1 × 10^7^, 1× 10^8^ and 2.5 × 10^8^ pfu of the COH04S1 vaccine candidate. In the context of lentiviruses (LV), a phase I/II clinical trial in 100 healthy volunteers is planned for a minigene-based LV vaccine containing multiple conserved SARS-CoV-2 regions [83]. Recruiting volunteers is in progress and the plan is to evaluate safety and immunogenicity after subcutaneous injection of 5 × 10^6^ dendritic cells (DCs) transduced with the LV-DC vector, combined with intravenous administration of 1 × 10^8^ antigen-specific CTLs.

Among self-amplifying RNA viruses, MV vectors have been utilized as vaccine candidates [50]. The evaluation of safety, tolerability and immunogenicity was planned for 90 volunteers to be vaccinated with the MV-SARS-CoV-2 vaccine candidate TMV-083 in a randomized, placebo-controlled, two-center phase I clinical study [89]. Initial findings from the phase I study were, however, disappointing, with weaker immune response in immunized volunteers than seen in COVID-19 patients, which led to the termination of the trial [84]. In another approach, the replication-competent VSV SARS-CoV-2 S vaccine candidate V590 was subjected to a phase I trial for safety and tolerability evaluation in 252 participants [90]. However, the highly efficacious use of VSV-based vaccines against Ebola virus did not generate similar success for V590 [85]. Although intramuscular administration of the vaccine candidate in the phase I trial demonstrated good tolerability, the immune responses were inferior to those observed in COVID-19 patients and the study was discontinued [85]. The VSV-ΔG vaccine, based on a replication competent VSV vector with the SARS-CoV-2 S protein replacing the VSV G protein, will be administered to 18–55-year-old volunteers in a phase I/II dose-escalation study [86]. Participants will receive a single dose of 5 × 10^5^, 5 × 10^6^ and 5 × 10^7^ pfu of VSV-ΔG, respectively, in the first part of the study. In the second part, elderly subjects will receive the same single injection as in part one or two injections of 5 × 10^5^ pfu 28 days apart.

Finally, as SARS-CoV-2 infections occur intranasally, a potential option for COVID-19 vaccine delivery comprises intranasal administration [91]. In this context, a phase I clinical trial has been registered in China for administration of an influenza virus vector expressing the SARS-CoV-2 S RBD (DelNS1-2019-nCoV-RBD-OPT1) as an intranasal spray [87]. The same vaccine candidate has also been registered for a phase II trial [88].

## 5. Conclusions

An impressive spectrum of viral vectors has been subjected to vaccine development against COVID-19, as described in preclinical studies in animal models (Table 1) and in clinical trials (Table 2). Among the 162 current preclinical studies, 35 are based on viral vectors or VLPs, and in the 52 clinical trials conducted or in progress, 14 have utilized viral vectors [38]. Clearly, adenovirus-based vectors have represented the most preferred approach, eliciting robust antibody responses and protection against SARS-CoV-2 challenges in both rodents and primates. Moreover, protection has also been achieved after immunization with LV, MVA, NDV and VSV vectors in rodents. The classic delivery route comprises intramuscular administration, but also intranasal sprays have shown promise for adenovirus, LV and influenza virus vectors.

In the context of clinical evaluation, most other viral vectors have only been subjected to phase I trials, so far. In contrast, several adenovirus-based vaccine candidates have successfully passed through phase I and II clinical studies with the most advanced vaccine candidates already evaluated in tens of thousands of volunteers. Although the phase III trial on the ChAdOx1 nCoV-19 vaccine candidate was temporarily placed on hold in September 2020 due to some suspect adverse events in volunteers [92], it resumed in the UK shortly thereafter and also on 23 October 2020 in the US [93]. Encouragingly, the ChAdOx1 nCoV-19 was approved in the UK at the end of December, with vaccinations starting in early January 2021. Other adenovirus-based vaccine candidates have also demonstrated promising preliminary results in late phase clinical trials. On another note, the Sputnik V vaccine was approved, based on preliminary phase I-II data in Russia already on 11 August 2020 prematurely without safety and efficacy data from phase III according to the WHO guidelines. Since then, phase III data from 19,000 individuals indicated a 91.4% efficacy rate [94], which has supported the approval of Sputnik V in Hungary, although not by the European Union’s medicines regulators, and in the United Arab Emirates.

Vaccine development against SARS-CoV-2, as well as against other infectious diseases, has demonstrated that prime-boost strategies can provide superior immunogenicity and protection against challenges with lethal doses of infectious agents. Typically, the efficacy of both nucleic acid- and virus-based COVID-19 vaccines have profited from sequential immunization protocols [33]. In this context, the prime-boost regimen for the Sputnik V vaccine candidate applied the adenovirus types 26 and 5 sequentially, not only to enhance the immune response compared to a single immunization, but also to overcome potential pre-existing immunity [42,70]. The mixing and matching of COVID-19 vaccine candidates have further been extended to the combination of the ChAdOx1 nCoV-19 and Sputnik V vaccines [95]. Moreover, a heterologous vaccination strategy applying a self-amplifying RNA and the ChAdOx1 nCoV-19 vaccine showed domination of cytotoxic T cells and Th1+ CD4 T cells superior to homologous vaccination regimens in mice [96]. Moreover, the prime-boost administration of ChAdOx1 nCoV-19 and the lipid nanoparticle-encapsulated mRNA-based vaccine BNT162b2 has been planned for a clinical trial [97].

Although not applied as viral vectors, self-amplifying alphavirus RNA vectors have been used as lipid nanoparticle-RNA formulations for vaccine development, eliciting dose-dependent neutralizing antibody responses in immunized mice, which was of a higher magnitude compared to levels obtained from recovered COVID-19 patients [53]. Moreover, the immunization of mice induced Th1-biased immune responses without triggering any ADE. Recently, a phase I trial in 18–45 years old volunteers has been initiated for safety and dose-escalation evaluations [98]. The LNP-saRNA vaccine candidate will be administered intramuscularly at doses of 0.1, 0.3 and 1.0 µg, and individuals will be followed up to one year.

In addition to virus-based vaccine candidates, numerous approaches of vaccine development based on inactivated or attenuated virus, protein subunits and peptides, and nucleic acids have taken place [4]. Although adenovirus-based COVID-19 vaccine candidates have demonstrated comparable efficacy to approved RNA-based vaccines, providing over 90% protection certain concerns have been raised. Despite their superiority related to delivery, any clinical use of viral vectors needs to be validated to guarantee safe delivery, which naturally is more demanding than administration of purified proteins, peptides or nucleic acids. Another concern relates to potential pre-existing immunity against viruses. Particularly, most of the current vaccine candidates are subjected to prime-boost immunization strategies, with the exception of the Ad26.COV2.S showing efficacy after a single vaccination [63,64,65,66]. For this reason, in one approach, a chimpanzee adenovirus vector with no pre-existing immunity was employed [72,74]. Another strategy comprises of using the Ad26 serotype for the prime immunization and the Ad5 serotype for the boost immunization to maximize the effect and to avoid pre-existing immunity [42,70]. Viral vectors have been suggested to cause more frequent and severe adverse events, and less flexible for modifications related to targeting new SARS-CoV-2 variants. On the other hand, virus-based vaccines are more stable than RNA-based vaccines, which significantly facilitates storage and transportation logistics for vaccine distribution. As described above, several studies on heterologous vaccination strategies combining RNA- and virus-based vaccines are in progress, profiting from the favorable features of each approach.

In summary, RNA- and viral vector-based COVID-19 vaccines have seen an unprecedented development related to speed and with a protection efficacy of more than 90%. As several viral vector-based vaccines will soon be on the market, the potential of overcoming the COVID-19 pandemic has improved and has also generated the means to be better prepared for potential future epidemics and pandemics. In this context, the recent findings of two new SARS-CoV-2 lineages containing the N150Y mutation in the RBD of the S protein spreading rapidly in the UK, due to a 10% higher transmission rates have raised some concerns [99]. Moreover, the combination of the N510Y and deletion of amino acids 69 and 70 in the S protein have turned out to be 75% more transmissible than the 501N lineage [100]. This British variant, B1.1.7, has now spread to more than 30 countries including the US [101]. The key question is whether current COVID-19 vaccines protect vaccinated individuals from these novel SARS-CoV-2 mutants. A recent study in 20 volunteers who received either of the two approved RNA vaccines showed a small but significant reduction in neutralizing antibody activity against the N501A or the K417N-E484K-N501Y combination [102]. In the context of the UK variant SARS-CoV-2 B.1.1.7, a phase II/III immunization study in volunteers with the ChAdOx1 nCoV-19 vaccine resulted in similar efficacy against B.1.1.7 as against other lineages [80]. In another study, the SARS-CoV-2 S protein nanoparticle vaccine produced in insect cells from a baculovirus vector showed 86% efficacy against the UK variant and 60% efficacy against the South African variant according to interim results from a phase III clinical trial [103].

In another study, isogenic N501 and Y501 SARS-CoV-2 were tested with sera from 20 individuals previously vaccinated with the BNT162b RNA vaccine, which showed equivalent neutralizing titers to the N501 and Y501 viruses [104]. In any case, as RNA viruses such as coronaviruses have a strong tendency to frequently mutate, the scientific community needs to remain alert and act properly to ensure that the current vaccines provide protection against novel mutant variations, and if that is not the case, rapidly engineer new vaccines targeting emerging mutations.

## Figures and Tables

**Figure 1 viruses-13-00317-f001:**
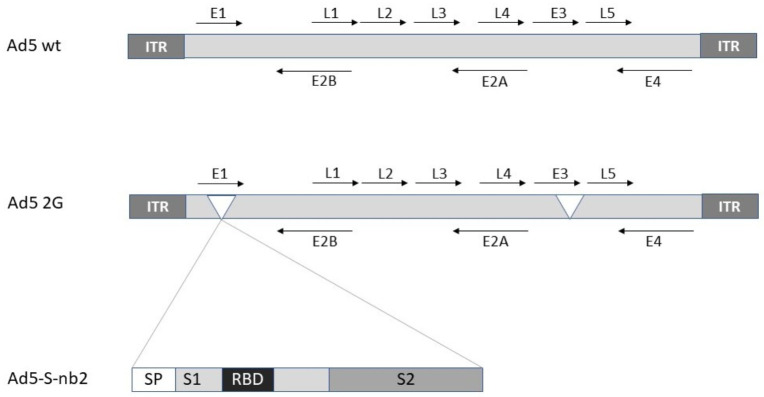
Adenovirus expression vector engineering for SARS-CoV-2 S expression. Ad5 wt, adenovirus type 5 wild type; Ad5 2G, second generation Ad5 vector with deletions in E1 and E3 genes; Ad5-S-nb2, Ad5-based vector expressing the SARS-CoV-2 S protein. E, early genes; L, late genes; RBD, receptor binding domain; S1 and S2, regions of S protein.

**Figure 2 viruses-13-00317-f002:**
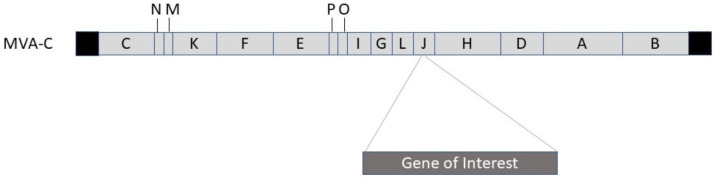
Vaccinia virus (MVA-C) expression vector engineered for recombinant protein expression. Structural MVA genes A, B, C, D, E, F, G, H, I, J, K, L, M, N, O, and P indicated. The gene of interest is introduced into the J gene.

**Figure 3 viruses-13-00317-f003:**
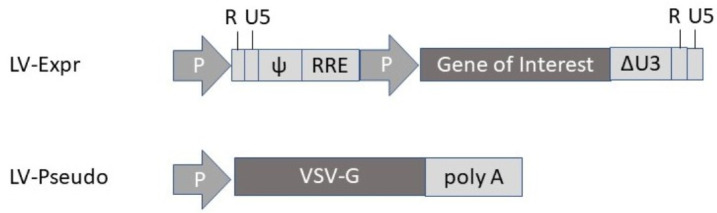
Lentivirus expression vector (LV-Expr) and envelope-expressing vector for production of pseudotyped lentivirus (LV-Pseudo) particles with vesicular stomatitis virus glycoprotein (VSV-G) envelope. P, promoter; ψ, packaging signal; RRE, Rev response element; ΔU3, deletion.

**Figure 4 viruses-13-00317-f004:**
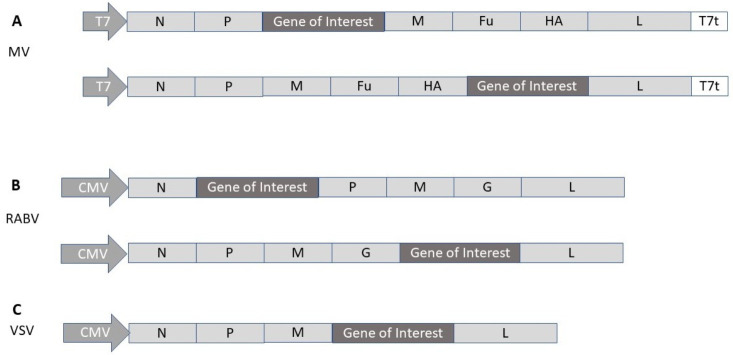
Measles virus (MV) and rhabdovirus expression vectors. (**A**) MV vector, where the gene of interest can be inserted either between the P and M genes or the HA and L genes. (**B**) Rabies virus (RABV) vector with the gene of interest inserted either between the N and P genes or the G and L genes. (**C**) Vesicular stomatitis virus (VSV) vector with the gene of interest inserted between the M and L genes. CMV, cytomegalovirus promoter; Fu, fusion protein; G, glycoprotein; HA, hemagglutinin; L, large; M, matrix; N, nucleocapsid protein; P, phosphoprotein; T7, T7 RNA polymerase promoter; T7t, T7 terminator.

**Figure 5 viruses-13-00317-f005:**
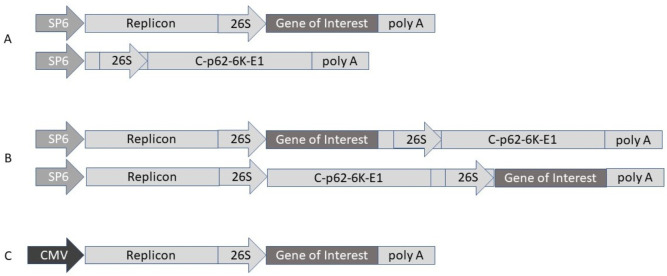
Alphavirus expression vectors. (**A**) Replication-deficient alphavirus system with expression vector (above) and helper vector (below). (**B**) Replication-proficient alphavirus system including two alternative insertion sites. (**C**) DNA/RNA layered Semliki Forest virus (SFV) vector. 26S, SFV 26S subgenomic promoter; CMV, cytomegalovirus promoter; Poly A, polyadenylation signal; Replicon, replicase (non-structural genes); SP6, SP6 RNA polymerase promoter.

**Table 1 viruses-13-00317-t001:** Examples of preclinical studies for viral vector-based SARS-CoV-2 vaccine candidates.

Vector	Construct	Response	Ref
**Adenovirus**			
Ad5	Ad5-S-nb2	Protection against SARS-CoV-2 in macaques	[7]
Ad5	Ad5-S-RBD	Systemic & mucosal response in mice (i.n. administration)	[39]
Ad26	Ad26.COV.S	Protection against SARS-CoV-2 pneumonia in hamsters	[40]
Ad26	Ad26.COV.S	Protection against SARS-CoV-2 in macaques	[41]
Ad26/Ad5	rAd26-S/Ad5-S	Good preclinical safety and immunogenicity profiles	[42]
ChAdOx1	ChAdOx1 nCOV-19	Protection against SARS-CoV-2 in macaques	[43]
ChAdOx1	ChAd-SARS-CoV-2-S	Protection of mice after i.n. administration	[44]
**Lentivirus**			
LV	LV-SARS-CoV-2-S	Protection in hamsters after i.n. administration	[45]
**Influenza**			
IFV	ΔNA(RBD)-Flu	Robust antibody responses after i.n. administration	[47]
**Poxvirus**			
MVA	MVA SARS-CoV-2-S/N	Humoral & cellular immune responses in mice	[46]
MVA	MVA-COV2-S	Protection against SARS-CoV-2 in mice	[47]
**NDV**			
NDV	NDV SARS-CoV-2-S	Protection against SARS-CoV-2 in mice and hamsters	[48]
NDV	NDV-SARS-CoV-2-S	Protection against SARS-CoV-2 in mice	[49]
**Measles virus**			
MV	MV-SARS-CoV-2-S	Neutralizing & T cell antibody responses in mice	[50]
**Rhabdovirus**			
VSV	VSV-SARS-CoV-2-S	Protection against SARS-CoV-2 in mice	[51]
VSV	VSV-Δ	Protection against SARS-CoV-2 in hamsters	[52]
**Alphavirus**			
VEE	VEE-SARS-CoV-2-S	High-level neutralizing antibodies in mice	[53]
**VLPs**			
eVLPs	MLV Gag SARS-CoV-2 S	Immunogenicity and efficacy in hamsters	[54]

eVLPs, virus-like particles; IFV, influenza virus; i.n., intranasal; LV, lentivirus; MLV, murine leukemia virus; MV, measles virus; MVA, modified vaccinia Ankara; NDV, Newcastle disease virus; RBD, receptor-binding domain. VSV, vesicular stomatitis virus; VEE, Venezuelan equine encephalitis virus; VSV-ΔG, VSV vector where the VSV-G protein is replaced by SARS-CoV-2 S protein.

**Table 2 viruses-13-00317-t002:** Examples of clinical studies for viral vector-based SARS-CoV-2 vaccine candidates.

Vector	Phase	Response	Ref
Ad5-SARS-CoV-2-S	I	SARS-CoV-2-specific cellular and humoral responses	[59]
Ad5-SARS-CoV-2-S	II	Strong neutralizing SARS-CoV-2-specific antibodies	[60]
Ad5-SARS-CoV-2-S	III	Recruitment in progress	[61]
Ad5-SARS-CoV-2-S	III	Study in progress	[62]
Ad26.COV2 S	I/II	Good safety, strong immunogenicity	[63,64]
Ad26.COV2 S	III	Recruitment in progress	[65]
Ad26.COV2 S	III	Recruitment in progress	[66]
rAd26-S/rAd5-S	I/II	Good safety, strong immune responses	[42]
rAd26-S/rAd5-S	III	Recruitment in progress	[67]
rAd26-S/rAd5-S	III	Study in progress	[68]
rAd26-S/rAd5-S	III	Study planned	[69]
rAd26-S/rAd5-S	III	91.6% vaccine efficacy from interim results	[70]
ChAdOx1 nCOV-19	I/II	Humoral and cellular immune responses	[72]
ChAdOx1 nCOV-19	II/III	Similar nAb responses in all age groups	[74]
ChAdOx1 nCoV-19	III	Recruitment in progress	[75]
ChAdOx1 nCoV-19	III	Recruitment in progress	[76]
ChAdOx1 nCoV-19	III	Study in progress	[77]
ChAdOx1 nCoV-19	III	Study in progress	[78]
ChAdOx1 nCoV-19	III	Interim results: 62.1–90.0% efficacy in 4 trials	[80]
MVA-SARS-COV-2	I	Recruitment in progress	[81]
MVA-SARS-COV-2	I	Recruitment in progress	[82]
LV-DC + CTL Ag	I	Recruitment in progress	[83]
MV-SASR-CoV-2-S	I	Trial discontinued	[84]
VSV	I	Trial discontinued	[85]
VSV	I/II	Recruitment in progress	[86]
IFV-CoV-2 S RBD	I	Registered trial	[87]
IFV-CoV-2 S RBD	II	Registered trial	[88]

Ad, adenovirus; Ag, antigen; ChAdOx1-S, simian adenovirus expressing SARS-CoV-2 S protein; CTLs, cytotoxic T lymphocytes; IFV, Influenza virus; LV-DCs, lentivirus-transduced dendritic cells; MV, measles virus; MVA, modified vaccinia virus Ankara; nAb, neutralizing antibody; RBD, receptor binding domain; VSV, vesicular stomatitis virus.

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
