# Peer review of "Viral Vectors for COVID-19 Vaccine Development"

_viruses, 2021, doi:10.3390/v13020317_

Round 1

Reviewer 1 Report

This is an excellent review of the current state-of-the-art with respect to use of various viral vectors for development of COVID-19 vaccines. 

The paper is an excellent and timely contribution to the literature. I notice on review of the paper a few minor points that merit consideration before publication.

  1. In Figure 2, there appears to be some discontinuity of the lines in the diagram for the N, M and H genes. The lines in other words are not straight. This should be checked.
  2. Logunov and colleagues recently reported interim results from a phase 3 trial of the Sputnik V COVID-19 vaccine in Lancet. The trial results show a consistent strong protective effect across all participant age groups. Also known as Gam-COVID-Vac, the vaccine uses a heterologous recombinant adenovirus approach using adenovirus 26 (Ad26) and adenovirus 5 (Ad5) as vectors for the expression of the severe acute respiratory syndrome coronavirus 2 (SARS-CoV-2) spike protein. The use of two varying serotypes, which are given 21 days apart, is intended to overcome any pre-existing adenovirus immunity in the population. This sequential approach with different adenovirus vectors to avoid anti-vector immunity should be commented on. The author may also comment on what the implications would be for a mixed schedule in which the priming dose was followed by another vaccine with a different platform. There is considerable discussion about "mixing and matching" different prime-boost approaches that might, for practical reasons or inadvertent reasons, use mixed platforms. Would this impair immune responses or protection?
  3. Table 2 does not define the abbreviation for IFV-CoV-2-S RBD vaccine. What does IFV stand for? This abbreviation should be provided in the table.
  4. The VBI platform, although not exactly a viral vector, does use plasmids expressing viral vector proteins expressing VLPs. The lead candidate, VBI-2901 for SARS-CoV-2, produces eVLPs after transient transfection of cells with plasmids encoding the murine leukemia virus (“MLV”) Gag and target surface or internal proteins of interest. MLV Gag expression induces “budding” of particles from membrane of transfected cells; the COVID protein is then incorporated. This is similar enough to a viral vectored vaccine that a mention of this approach should be included in the review and tables.
  5. Merck recently announced announced that the company is discontinuing development of its SARS-CoV-2 vaccine candidates, V590 and V591. The V590 vaccine was based on a vesicular stomatitis virus vector-based platform and the V591 was based on a measles platform. The author should provide a review of these vectored platforms. Specifically, why did these vectors fail when others, such as adenovirus vectors, succeeded? Is there something intrinsically less immunogenic or a problem with processing or presentation of antigen using the VSV or measles vectors? For this review, help the reader understand and put into perspective these questions.
  6. It would be valuable to have a supplemental table providing links to the clinicaltrials.gov numbers describing the vectored COVID vaccines that have gone into, or are actively being studied, in clinical trials.

Author Response

  1. In Figure 2, there appears to be some discontinuity of the lines in the diagram for the N, M and H genes. The lines in other words are not straight. This should be checked.

R: Figure has been revised.

  1. Logunov and colleagues recently reported interim results from a phase 3 trial of the Sputnik V COVID-19 vaccine in Lancet. The trial results show a consistent strong protective effect across all participant age groups. Also known as Gam-COVID-Vac, the vaccine uses a heterologous recombinant adenovirus approach using adenovirus 26 (Ad26) and adenovirus 5 (Ad5) as vectors for the expression of the severe acute respiratory syndrome coronavirus 2 (SARS-CoV-2) spike protein. The use of two varying serotypes, which are given 21 days apart, is intended to overcome any pre-existing adenovirus immunity in the population. This sequential approach with different adenovirus vectors to avoid anti-vector immunity should be commented on.

R: Text has been added to stress the point of using two different adenovirus serotypes to overcome pre-existing immunity against adenoviruses. Information on the interim results from the Phase III trial (published a week ago!) has been added to the text, Table 2 and the references.

The author may also comment on what the implications would be for a mixed schedule in which the priming dose was followed by another vaccine with a different platform. There is considerable discussion about "mixing and matching" different prime-boost approaches that might, for practical reasons or inadvertent reasons, use mixed platforms. Would this impair immune responses or protection?

R: A paragraph has been added to the Conclusions section on prime-boost strategies including mixing and matching.

  1. Table 2 does not define the abbreviation for IFV-CoV-2-S RBD vaccine. What does IFV stand for? This abbreviation should be provided in the table.

R: IFV stands for influenza virus, which has now been added to the footnote of Table 2.

  1. The VBI platform, although not exactly a viral vector, does use plasmids expressing viral vector proteins expressing VLPs. The lead candidate, VBI-2901 for SARS-CoV-2, produces eVLPs after transient transfection of cells with plasmids encoding the murine leukemia virus (“MLV”) Gag and target surface or internal proteins of interest. MLV Gag expression induces “budding” of particles from membrane of transfected cells; the COVID protein is then incorporated. This is similar enough to a viral vectored vaccine that a mention of this approach should be included in the review and tables.

R: Text has been added and eVLPs included in Table 1.

  1. Merck recently announced that the company is discontinuing development of its SARS-CoV-2 vaccine candidates, V590 and V591. The V590 vaccine was based on a vesicular stomatitis virus vector-based platform and the V591 was based on a measles platform. The author should provide a review of these vectored platforms. Specifically, why did these vectors fail when others, such as adenovirus vectors, succeeded? Is there something intrinsically less immunogenic or a problem with processing or presentation of antigen using the VSV or measles vectors? For this review, help the reader understand and put into perspective these questions.

R: Text has been added to address the discontinuation of the two trials.

  1. It would be valuable to have a supplemental table providing links to the clinicaltrials.gov numbers describing the vectored COVID vaccines that have gone into, or are actively being studied, in clinical trials.

R: I agree that a supplemental table could be considered. However, as the status of clinical trials on COVID-19 vaccines is so dynamic, changing frequently, the daily information is easily available in reference 39, which is the continuously updated WHO website at https://www.who.int/publications/m/item/draft-landscape-of-covid-19-candidate-vaccines

Reviewer 2 Report

As I read the manuscript I found it technically sound and very well written. It gives a very informative overview of the state of the art of using viral vectors for COVID-19 vaccine development.

Author Response

As I read the manuscript I found it technically sound and very well written. It gives a very informative overview of the state of the art of using viral vectors for COVID-19 vaccine development.

R: Thank you for the kind comments. I noticed that the reviewer suggested minor spell checking of the manuscript, which has been now done.

Reviewer 3 Report

Overview:

The author presents a brief review of viral vector-based vaccines for SARS-CoV-2, including adeno, lenti, vaccina and other less commonly utilized vectors.  The article is timely, and I believe it will be of interest to readers.  However, I find there are additions that are needed to better describe this research topic, as well as some other additions that might be more helpful to the general reader who is not fully familiar with viral vectored vaccines. I’ve listed these suggestions below:

General Comments:

1) Section 2.2:  I find the poxvirus/MVA intro paragraph to be a bit lacking compared to the descriptions given for adeno and lentiviruses, and especially considering the long history of this type of virus as a vaccine vector. Please consider expanding.

2) Section 2.3: The author may consider adding to this section that many groups are now using lenti-virus based, SARS-CoV-2 psuedoviruses as surrogates in COVID-19 sera neutralization assays.  While this isn’t exactly a vaccine, it shows how this important type of vector can be used to not only immunize, but analyze the immune response to SARS-CoV-2 protein antigen expression in humans and animal models. 

3) Table 1 would benefit from an extra column that lists the name of the virus/virus family (i.e. adenovirus; influenza virus, modified vaccinia, etc.) for quick abbreviation references for the reader.

4) Section 4.  This section needs the addition of the groups/organizations responsible for the specific trials mentioned.  This is specifically needed for the first paragraph.

5)  Line 258:  Updates on clinical data of the Russian adeno vaccine are now available and may benefit from inclusion here.

6) Lines 260-277:  This section needs to be updated as a great deal of information on the ChAdOx vaccine has become available, likely during the time this manuscript was in review.  In this fast moving situation, new efficacy data, as well as efficacy on SARS-CoV-2 variants and neutralization by ChAdOx convalescent sera have been reported.  Please add this information here.

7) Table 2:  This table would benefit from a new column that shows the organization/institution that is developing this vaccine.  Similar to the WHO draft table on COVID vaccine development (https://www.who.int/publications/m/item/draft-landscape-of-covid-19-candidate-vaccines).

8)  A major discrepancy of this review is the failure to describe the problem of pre-existing immunity/immunity to the vector itself as a problem with viral vectored  vaccines.  This absolutely needs to be address in a new section and the implications of this on covid-vaccination should the virus become seasonally endemic are critically important. Further, some military personal in the US are vaccinated against Adenovirus and may have prexisting immunity.  The use of the ChAdOx vaccine may circumvent situations like this, but how does affect other adenovectors?

9)  A new paragraph, perhaps towards the end, that generally compares the benefits/negatives of viral vectored (or just adeno since they have progressed the furthest) competitors preparing to enter the clinic (Phizer/moderna mRNA, Novavax subunit, Sinovac/Sinpharm whole-inactivated, Inovio DNA, etc.) would be beneficial.  A general focus may be on the fact that ‘protection’ rates of adeno vaccines (at least ChAdOx) appear lower than for mRNA and recomb subunit vaccines, and rates of adverse effects appear more severe or common (with J&J – which was apparently critical enough to drop the boost).  Combined with immune responses to the vector, what does this say about the future of viral vectored vaccine for clinical use?

Minor Comments:

Line 119:  The phrasing in “have previously been frequently utilized” reads awkwardly.  Considering rewording.

Line 131:  Please reword “carried out” to avoid a dangling preposition.   

Line 147: ‘administration’ should not be capitalized

Line 150:  Suggest changing ‘gave’ to ‘conferred’

Line 154:  Suggest making a new paragraph at, ‘A COVID-19 vaccine candidate…’

Line 161:  Suggest making a new paragraph at, ‘The full length SARS-CoV-2 S protein…’

Line 174:  Suggest changing to ‘mice expressing ACE2’ or ‘ACE2 humanized mice’

Line 188:  This is a vague statement.  What does immunity in the lung mean specifically?

Author Response

The author presents a brief review of viral vector-based vaccines for SARS-CoV-2, including adeno, lenti, vaccina and other less commonly utilized vectors.  The article is timely, and I believe it will be of interest to readers.  However, I find there are additions that are needed to better describe this research topic, as well as some other additions that might be more helpful to the general reader who is not fully familiar with viral vectored vaccines. I’ve listed these suggestions below:

General Comments:

1) Section 2.2:  I find the poxvirus/MVA intro paragraph to be a bit lacking compared to the descriptions given for adeno and lentiviruses, and especially considering the long history of this type of virus as a vaccine vector. Please consider expanding.

R: Additional text and reference 12 has been added to section 2.2. An introduction to MVA has also been added to section 3.

2) Section 2.3: The author may consider adding to this section that many groups are now using lenti-virus based, SARS-CoV-2 psuedoviruses as surrogates in COVID-19 sera neutralization assays.  While this isn’t exactly a vaccine, it shows how this important type of vector can be used to not only immunize, but analyze the immune response to SARS-CoV-2 protein antigen expression in humans and animal models.

R: I disagree respectfully. Although lentivirus-based pseudoviruses are indeed useful tools for analysis of antigen expression, the focus here is on vaccines and it would not serve the purpose of the manuscript. 

3) Table 1 would benefit from an extra column that lists the name of the virus/virus family (i.e. adenovirus; influenza virus, modified vaccinia, etc.) for quick abbreviation references for the reader.

R: An additional column would make it difficult to fit Table 1 in the text, so instead subheadings of virus families have been inserted in the first column.

4) Section 4.  This section needs the addition of the groups/organizations responsible for the specific trials mentioned.  This is specifically needed for the first paragraph.

R: I disagree respectfully of including this information in a scientific review as it can easily be retrieved from reference 39.

5)  Line 258:  Updates on clinical data of the Russian adeno vaccine are now available and may benefit from inclusion here.

R: Un update on new clinical data (published last week!) has been added.

6) Lines 260-277:  This section needs to be updated as a great deal of information on the ChAdOx vaccine has become available, likely during the time this manuscript was in review.  In this fast moving situation, new efficacy data, as well as efficacy on SARS-CoV-2 variants and neutralization by ChAdOx convalescent sera have been reported.  Please add this information here.

R: Indeed, the field is fast-moving, and it is difficult to include the latest data. However, some updates have been included.

7) Table 2:  This table would benefit from a new column that shows the organization/institution that is developing this vaccine.  Similar to the WHO draft table on COVID vaccine development (https://www.who.int/publications/m/item/draft-landscape-of-covid-19-candidate-vaccines).

R: The same problem as for Table 1 with addition of another column and as the reviewer indicates this information is available from the WHO (reference 38).

8)  A major discrepancy of this review is the failure to describe the problem of pre-existing immunity/immunity to the vector itself as a problem with viral vectored vaccines.  This absolutely needs to be address in a new section and the implications of this on covid-vaccination should the virus become seasonally endemic are critically important. Further, some military personal in the US are vaccinated against Adenovirus and may have prexisting immunity.  The use of the ChAdOx vaccine may circumvent situations like this, but how does affect other adenovectors?

R: Text has been added on the use of different adenovirus serotypes to address this issue.

9)  A new paragraph, perhaps towards the end, that generally compares the benefits/negatives of viral vectored (or just adeno since they have progressed the furthest) competitors preparing to enter the clinic (Phizer/moderna mRNA, Novavax subunit, Sinovac/Sinpharm whole-inactivated, Inovio DNA, etc.) would be beneficial.  A general focus may be on the fact that ‘protection’ rates of adeno vaccines (at least ChAdOx) appear lower than for mRNA and recomb subunit vaccines, and rates of adverse effects appear more severe or common (with J&J – which was apparently critical enough to drop the boost).  Combined with immune responses to the vector, what does this say about the future of viral vectored vaccine for clinical use?

R: Text has been added to the Conclusions section comparing virus-based vaccines to other types of vaccines highlighting the advantages and disadvantages.

Minor Comments:

Line 119:  The phrasing in “have previously been frequently utilized” reads awkwardly.  Considering rewording.

R: Revised to “have previously been frequently applied”

Line 131:  Please reword “carried out” to avoid a dangling preposition.

R: “carried out” has been replaced by “conducted”   

Line 147: ‘administration’ should not be capitalized

R: Correction has been made.

Line 150:  Suggest changing ‘gave’ to ‘conferred’

R: Correction has been made.

Line 154:  Suggest making a new paragraph at, ‘A COVID-19 vaccine candidate…’

R: Correction has been made.

Line 161:  Suggest making a new paragraph at, ‘The full length SARS-CoV-2 S protein…’

R: Correction has been made.

Line 174:  Suggest changing to ‘mice expressing ACE2’ or ‘ACE2 humanized mice’

R: Correction has been made.

Line 188:  This is a vague statement.  What does immunity in the lung mean specifically?

R: The text has been revised accordingly.

Round 2

Reviewer 3 Report

The author has addressed the major concerns with the initial round of review.  Please consider adding citations for the MVA-IAV and MVA-Ebola studies in the new text added at Lines 57-62. My suggestion for a column heading in Table 2 remains.  However, neither suggestion precludes publication in current form.